# Doppler-defined pulmonary hypertension in β-thalassemia major in Kurdistan, Iraq

Ameen M. Mohammad[ID]¹☯*, Mohammed M. Dawad²‡, Muna A. Kashmoola³‡, Nasir Al-Allawi⁴☯

**1** Department of Internal Medicine, College of Medicine, University of Duhok, Duhok, Iraq, **2** Department of Hematology, Alkhansa Teaching Hospital, Mousel, Iraq, **3** Department of Pathology, College of Medicine, University of Mousel, Mousel, Iraq, **4** Department of Pathology, College of Medicine, University of Duhok, Duhok, Iraq

☯ These authors contributed equally to this work.
‡ These authors also contributed equally to this work.
* doctoramb@yahoo.com

**Data Availability Statement:** All relevant data are within the manuscript and its Supporting Information files.

**Funding:** The author(s) received no specific funding for this work.

## Abstract

Cardiopulmonary complications are among the most important complications of thalassemia major. Pulmonary hypertension is among these complications and studies addressing its frequency and associations in the latter disorder are sparse from Iraq. For this purpose a total 100 thalassemia major patients (≥ 8 years old) were enrolled from a main thalassemia center in Kurdistan, Northern Iraq. All patients had a full history and clinical examination. Full blood count, biochemical tests and viral screen including hepatitis B surface antigen and hepatitis C virus antibody, in addition to transthoracic Doppler echocardiography for tricuspid regurgitation jet velocity (TRV). The enrolled patients had a mean (SD) age of 17.6 (5.5) years, and included 52 males and 48 females. Pulmonary hypertension as defined by TRV> 2.8 m/s coupled with both exertional dyspnea and an absence of left sided heart failure, was identified in nine patients (9%). The latter subgroup of patients had significantly higher reticulocyte counts, S. LDH, S. ferritin, and hepatitis C sero-positivity compared to those without this complication by univariate analysis. While by multivariate logistic regression only reticulocytes and hepatitis C sero-positivity remained significant. Furthermore, TRV as a continuous variable was positively correlated with reticulocytes, S. bilirubin and LDH ($p<0.001$, $p = 0.002$ and $p<0.001$ respectively), but not with age or S. ferritin ($p = 0.77$, and $p = 0.93$ respectively). In conclusion, pulmonary hypertension is not uncommon in Iraqi patients with thalassemia major, and it appears to be linked to chronic hemolysis rather than iron overload.

## Introduction

β-thalassemia is an autosomal recessive disorder associated with defective synthesis of β-globin chains of hemoglobin [1]. It is not only common in the Mediterranean region, South-East Asia, the Indian subcontinent and the Middle East but has now become a global problem, spreading through migration worldwide [2, 3]. β-thalassemia is associated with a variety of

**Competing interests:** The authors have declared that no competing interests exist

phenotypes ranging from severe transfusion- dependent beta thalassemia major (TM) to asymptomatic β-thalassemia minor. Between these two extremes lies thalassemia intermedia (TI), which is a less severe phenotype than TM where most patients would only require sporadic or no transfusions; however, it is associated with several more frequent complications including increased thrombotic tendency and pulmonary hypertension (PH) [4]. The development of pulmonary hypertension in thalassemia may be the result of a multitude of pathogenic mechanisms, including chronic hemolysis, iron overload and hypercoagulability. One important consequence of chronic hemolysis and erythrocyte dysfunction is abnormal arginine-nitric oxide bioavailability [5, 6]. The latter may contribute to activation of platelets, endothelial dysfunction and increase oxidative stress leading to tissue damage throughout the cardiovascular system [7].

Previous studies from Iraqi Kurdistan have looked at the frequency and associations of pulmonary hypertension in patients with sickle cell anemia and thalassemia intermedia [8, 9], while reports on PH in TM are sparse. Accordingly, this study was initiated aiming at determining the frequency, correlations and the predictive factors of PH in a cohort of thalassemia major patients attending a main thalassemia center in this part of the country.

## Patients and methods

### Patients

A total of 100 consecutive TM patients who were eight years or older, visiting Jin Pediatric Thalassemia Center in Duhok, Kurdistan, Iraq during the period between September 2015 and February 2016, were recruited prior to their next scheduled transfusion.

### Clinical and laboratory assessment

The patients were clinically re-evaluated, and their records rechecked at the time of enrollment. Their current treatment, including chelation therapy, was carefully reviewed. Blood samples were taken for the following hematological and biochemical, investigations: complete blood count (CBC), reticulocyte (%), liver and renal function tests, and serum lactic dehydrogenase (LDH) using standard laboratory procedures. Serum ferritin, hepatitis B surface antigen (HBsAg) and hepatitis C antibody were assessed using a second-generation Cobas c501 Biochemistry auto-analyzer (Roche, HITACHI, Japan).

### Transthoracic echocardiography

Standardized transthoracic echocardiographic (TTE) examination and Doppler study were performed for all cases in the Echocardiography Laboratory of Azadi Teaching Hospital, Duhok, Iraq (Philips Hd5 Color Doppler Ultrasound Machine, Philips Medical Systems, Netherlands). Tricuspid valve regurgitation was measured in the apical four chambers, with parasternal short axis views, and a minimum of three sequential spectra. Pulmonary hypertension for the purposes of this study was defined as Tricuspid Regurgitant Jet Velocity (TRV) in excess of 2.8 m/s coupled with both exertional dyspnea, and an absence of left heart failure [10]. Furthermore, in addition to measuring the left ventricular ejection fraction (LVEF%), the diastolic function of the left ventricle was checked by pulsed Doppler-derived left ventricle diastolic mitral inflow in the apical four-chamber view, to measure the E and A peak velocities and their ratio (E/A). All echocardiographic measurements were done according to the American Society of Echocardiography recommendations [11]. Heart failure was diagnosed based on Framingham criteria [12].

## Ethical consideration

Informed consent was obtained from all participants or their guardians upon enrollment. The study was approved by the Ethics and Scientific Committee of the Iraqi Board of Medical Specialization, Baghdad, Iraq.

## Statistical analyses

All statistical analyses were performed using SPSS software (release 20; SPSS inc, Chicago, IL, USA). Fisher's exact test was used to compare categorical variables. Student t-test of independent samples was used for univariate analysis of continuous variables. Multivariate binary logistic regression was used for multivariate analysis. Pearson's correlation was used to assess the correlation between two continuous variables. All tests were two-sided, with a 0.05 level of significance.

## Results

The mean age of the enrolled patients was 17.6 years, with a standard deviation (SD) of 5.5 years, and included 52 males and 48 females (M:F ratio of 1.1:1). All patients were on regular transfusions, with a mean annual number of transfusions of 16.8 units (SD 2.3). Ninety two patients were on iron chelation therapy. The latter included 59 on oral deferasirox, 19 on subcutaneous deferoxamine, and 14 on combination of the two chelators. Sixty patients (60%) were splenectomized. Hepatomegaly was encountered in 16 patients (16%). The main laboratory characteristics at the time of enrollment are outlined in Table 1.

Pulmonary hypertension as defined by this study was found in 9 patients (9%) [including 3 patients whose TRV≥ 3.2 m/s]. Moreover, TRV was borderline (>2.5-≤2.8 m/sec) in another 16 (16%). Among PH group, reticulocyte (%), LDH, HCV sero-positivity, and serum ferritin were significantly higher than in the non-PH group by univariate analysis. Furthermore, those with PH had higher frequency of females and were more frequently splenectomized, though neither were significant (Table 2). Multivariate logistic regression to include those variables

**Table 1. Main laboratory findings among 100 thalassemia major patients at enrollment.**

| Parameter | Mean (SD) |
|---|---:|
| Hemoglobin (g/dL) | 8.6 (1.16) |
| WBC (x $10^9$/L) | 18.4 (14.5) |
| Platelets (x $10^9$/L) | 360 (201) |
| Reticulocyte (%) | 5.0 (2.2) |
| ALT (IU/L) | 73.7(52.1) |
| AST (IU/L) | 72.1 (51.3) |
| Bilirubin (mg/dL) | 1.67 (0.84) |
| Urea (mg/dl) | 30.6 (12.2) |
| Creatinine (mg/dl) | 0.49 (0.15) |
| Ferritin (ug/L) | 3903 (2641) |
| LDH (U/L) | 303.8 (150.1) |
| HBsAg | 5 (5%)* |
| HCV Antibody | 25 (25%)* |

Results are mean (standard deviation)

*number (percentage). WBC: white blood cell, ALT: serum alanine aminotransferase, AST: serum Aspartate aminotransferase, LDH: lactate dehyrdogenase, HBsAG: hepatitis B surface antigen, HCV: hepatitis C virus

**Table 2. Comparison between various parameters in patients with TRV> 2.8 and those with TRV ≤ 2.8 m/s.**

| Parameter | TRV>2.8 m/s | TRV≤ 2.8 m/s | Univariate |
|---|---|---|---|
| Age (years) | 18.6 ± 4.6 | 17.5±5.6 | 0.59 |
| Sex | 2 M/7 F | 50 M/41 F | 0.125 |
| Age of starting transfusions (months) | 8.1 ± 2.6 | 7.2 ± 2.4 | 0.296 |
| Annual rate of Transfusions | 18.1±1.9 | 16.6 ± 2.3 | 0.063 |
| Splenectomy | 7 (77.8%) | 53 (58.2%) | 0.439 |
| Hepatomegaly | 1 (11.1%) | 15 (16.5%) | >0.99 |
| Hemoglobin (g/dL) | 8.6 ±1.1 | 8.7± 1.2 | 0.85 |
| Leucocyte count (x10$^9$/L) | 22.2 ± 19.3 | 18.0 ± 14 | 0.41 |
| Platelets (x10$^9$/L) | 392 ± 219 | 357 ± 200 | 0.62 |
| Reticulocyte (%) | 7.52 ±1.1 | 4.77 ± 2.1 | <0.001 |
| ALT (IU/L) | 93.4 ± 41.6 | 71.8 ± 52.8 | 0.236 |
| AST (IU/L) | 85.1 ±36.1 | 70.8 ± 52.5 | 0.427 |
| Bilirubin (mg/dL) | 1.99 ±0.96 | 1.64 ± 0.82 | 0.236 |
| Urea (mg/dl) | 27.9 ±5.06 | 30.9 ± 12.64 | 0.480 |
| Creatinine (mg/dl) | 0.45 ± 0.15 | 0.50 ± 0.15 | 0.333 |
| S. Ferritin (ug/L) | 5603 ±3172 | 3735 ± 2542 | 0.042 |
| LDH (U/L) | 520.1 ±224.7 | 282.4 ± 123.1 | <0.001 |
| HCV sero-positivity | 8 (88.9%) | 17 (18.6%) | <0.001 |
| HBsAg positivity | 1 (11.1%) | 4 (4.4%) | 0.765 |
| LVEF (%) | 58.8 ±5.9 | 62.1 ± 6.0 | 0.117 |
| E/A | 1.71 ± 0.32 | 1.76 ± 0.24 | 0.557 |

WBC: white blood cell, ALT: alanine aminotransferase, AST: aspartate aminotransferase, LDH: lactate dehydrogenase, HBsAG: hepatitis B surface antigen, HCV: hepatitis C virus, LVEF: left ventricular ejection fraction, E/A: ratio for diastolic function of left ventricle, TRV: Tricuspid Regurgitant Jet velocity.

with a p ≤ 0.1, demonstrated that only reticulocyte count and HCV positivity remained significantly higher (Table 3). Left ventricular systolic dysfunction (as defined by LVEF% <50%) was identified in three patients in the non-PH group. All the three patients satisfied the Framingham criteria for diagnosis of heart failure.

When age at enrollment, age at starting transfusions, serum ferritin, reticulocyte count (%), serum bilirubin, LVEF%, E/A and serum LDH correlations with TRV as continuous variables were assessed using Pearson correlation, none except S. LDH, serum bilirubin and reticulocyte count (%) showed a significant positive correlation ($p<0.001$, $p = 0.002$, and $p<0.001$ respectively) (Figs 1 and 2; Table 4).

**Table 3. Multivariate analysis to compare between those with TRV >2.8 and those with TRV ≤ 2.8 m/s on parameters with a $p$ value ≤0.1 by univariate analysis.**

| Parameter | B | S.E. | Wald | $p$ value. | Odds Ratio | 95% Confidence Interval OR |
|---|---|---|---|---|---|---|
| Reticulocyte count | 0.709 | 0.347 | 4.169 | 0.041 | 2.031 | 1.029–4.010 |
| LDH | 0.003 | 0.003 | 1.333 | 0.248 | 1.003 | .998–1.009 |
| Ferritin | 0.000 | 0.000 | 1.383 | 0.240 | 1.000 | 1.000–1.001 |
| Annual Transfusion Frequency | 0.413 | 0.310 | 1.774 | 0.183 | 1.512 | .823–2.778 |
| HCV Positivity | 3.273 | 1.278 | 6.560 | 0.010 | 26.402 | 2.157–323.214 |

LDH: Lactate Dehydrogenase; HCV: Hepatitis C Virus.

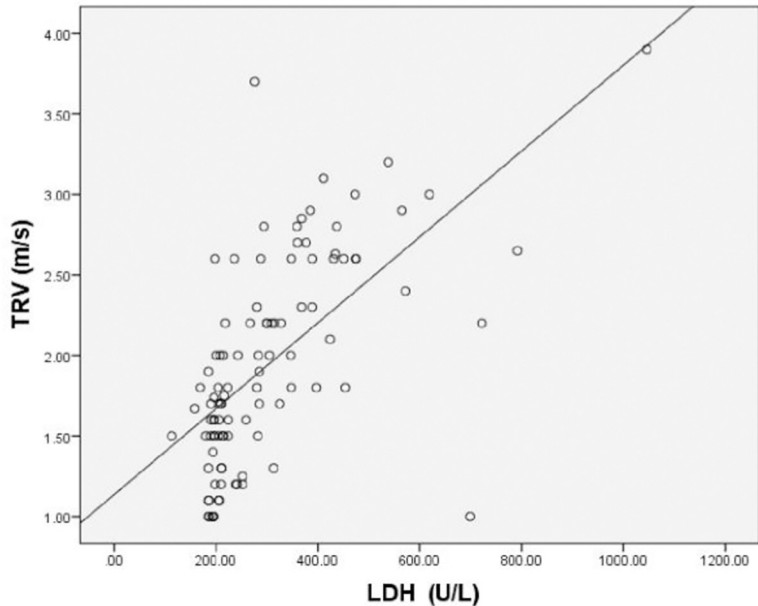

**Fig 1. Scatter plot showing the correlation between TRV and LDH in 100 thalassemia major patients.**

## Discussion

While PH is a well-recognized complication of TM, its doppler-defined frequencies in various TM series showed remarkable variability ranging from 1.7% to 79% [13–19]. The most important cause of such variability is the choice of the TVR cut-off point. While many studies used a cut- off of 2.5 m/s, others used 2.8 m/s, or higher values of 3.0 or 3.2 m/sec. The choice of a higher cut-off would ensure higher specificity, but on the expense of sensitivity [20]. Looking

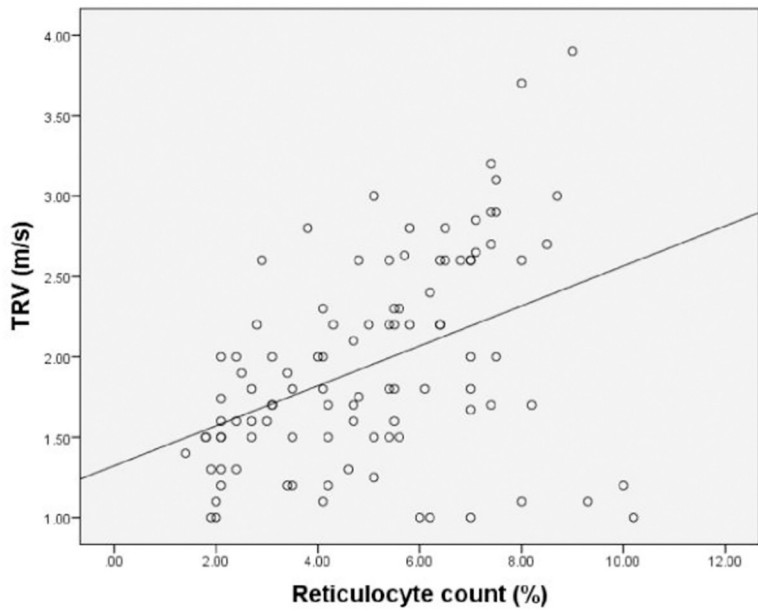

**Fig 2. Scatterplot showing correlation between TRV and reticulocyte count in 100 thalassemia major patients.**

Table 4. Correlation between continuous variables with TRV.

| Parameter | Pearson correlation coefficient | P-value |
|---|---|---|
| Age | 0.029 | 0.774 |
| Age at starting transfusions | -0.054 | 0.595 |
| Annual Transfusion frequency | 0.142 | 0.160 |
| Hemoglobin | -0.129 | 0.20 |
| Reticulocyte count | 0.419 | <0.001 |
| Bilirubin | 0.307 | 0.002 |
| LDH | 0.623 | <0.001 |
| Leucocyte count | -0.009 | 0.93 |
| Platelet count | 0.002 | 0.986 |
| S. Ferritin | 0.009 | 0.929 |
| Ejection fraction (LVEF%) | -0.153 | 0.129 |
| E/A | -0.018 | 0.855 |

LDH: lactate dehydrogenase, LVEF: left ventricular ejection fraction, E/A: ratio for diastolic function of left ventricle, TRV: Tricuspid Regurgitant Jet velocity.

at our data, 25% had TVRs>2.5m/s, compared to 9% and 3% for TVRs >2.8 m/s and ≥ 3.2 m/s respectively. For the purposes of the current study we used the cut-off point of 2.8 m/s, coupled by exertional dyspnea and in the absence of left ventricular failure, to increase specificity and decrease false positive cases [10, 21]. It is important to note that the gold standard for the diagnosis of PH is right heart catheterization (RHC), and it has been documented that despite moderate correlation between RHC and echocardiography, the latter tends to overestimate the actual frequency of PH [22]. However, RHC is an invasive procedure, thus the vast majority of reports, similar to ours, utilized echocardiography to screen for PH in hemoglobinopathies.

Another cause of variability in the frequency of PH in TM is the adequacy of management. Several investigators reported that in well-managed TM patients PH is virtually absent, while others reported high PH frequencies among TM series where management was inadequate [19, 23, 24]. Actually it has been proposed that timely transfusion and adequate chelation may delay the emergence of PH in TM patients [23, 24], by restoring tissue delivery of oxygen, suppressing production of defective red cells, reducing hemolysis and ameliorating hypercoagulability [25]. Among our patients, the management as judged by pre-transfusion hemoglobin and the mean serum ferritin seems to be less than optimal, since pre-transfusion hemoglobin was more than 9 g/dl in only 41%, compared to nearly 90% in some studies from developed countries, while our mean serum ferritin was evidently higher than reported from the latter countries[26, 27]. Another cause of variation in the reported rates of PH among TM patients is the age range of the recruited patients, and studies including mainly adults would be expected to have higher rates, than those focusing mainly on children. In this series we did not include children less than 8 years old, since PH is unlikely and we enrolled patients older than this age, with our youngest patient diagnosed as PH being 12 years old, which is consistent with the literature [17].

Pulmonary hypertension is more likely to occur among thalassemia intermedia (TI) than TM, and our rate of 9% is less than the 20.4% reported in an earlier study from our center on TI using the same diagnostic criteria for PH [9]. The lower frequency of PH among TM, compared to TI, has also been documented by many previous studies worldwide [18, 24, 28]. It appears that more frequent and regular transfusion in TM may be the main contributor to this difference. In the earlier study on TI at our center, 83.8% were sparsely or rarely transfused

[9]. This notion may be at the center of the argument of the need to transfuse patients with non-transfusion dependent thalassemia (NTDT) more regularly to avoid or delay PH [28].

Female predominance, though insignificant, among patients with PH in the current study is shared by some reports [29, 30], but disputed by others [17–19]. Strong female predominance in many forms of PH is well recognized [31]. This has been ascribed to altered estrogen levels and metabolism in the females, which affect the pulmonary vasculature and the right ventricle in a way that predisposes to PH [32].

A significant association of PH with hepatitis C sero-positivity as detected in the current study, is matched by some studies on TM [17]. This observation is not unexpected since HCV infection has been linked to right ventricular dysfunction, increased pulmonary vascular resistance and pulmonary hypertension [33]. Furthermore, chronic liver disease has been associated with pulmonary hypertension [34, 35].

Several studies have implicated splenectomy as a risk factor for PH in thalassemia [17, 18, 29, 36]. It appears that the loss of the spleen filtering functioning may lead to increased circulating abnormal red cells, nucleated red cells, red cell breakdown products, platelet activation and pro-coagulant factors release, which may eventually lead to increased risk of intravascular thrombosis and changes in pulmonary vasculature leading to PH [36–38]. In the current study, the proportion of splenectomized patients was much higher in the PH subgroup, but did not reach significance. Such absence of significant association is not unique and has been reported by at least one previous study [19].

The association between higher reticulocytes and LDH with PH, and between higher reticulocytes, LDH and bilirubin with TRV is interesting and likely indicates that PH (as identified by TRV >2.8 m/s) and increasing TRV (as a continuous variable regardless of any cut-off points) are linked to chronic hemolysis. Hemolysis would lead to release of cell-free hemoglobin and arginase and thus lead to impaired nitric oxide availability and endothelial dysfunction and eventually pulmonary hypertension, and this has documented in both sickle cell anemia and thalassemia [39, 40].

Although serum ferritin was significantly higher among PH patients by univariate analysis, this did not remain so in the multivariate one. Similarly, we failed to find an association between ferritin and TRV. This favors the conclusion that among our patients and in consistence with some earlier studies, chronic hemolysis, rather than iron overload, plays a major role in the pathogenesis of PH in our TM patients [29, 30, 41]. This is consistent with the notion that among the less than adequately transfused/chelated patients, hemolysis rather than iron overload is the main culprit of PH [5].

The importance of early identification of PH in thalassemia should be underscored, since it is associated with right ventricular dysfunction/failure and increased mortality [42]. While a variety of management options have been tried, ranging from the institution of more adequate transfusion and chelation, to hydroxycarbamide, L-carnitine, phosphodiesterase type 5 inhibitors, endothelial receptor antagonists and synthetic prostacyclins, however these therapeutic option are still waiting validation by double blind trials [5, 43].

## Strengths and limitations

The main limitation of the current study is that it did not include using the invasive RHC to confirm PH, though the use of TRV cut off point of 2.8 m/s coupled by clinical findings was shown by earlier studies to correlate well with RHC. Another limitation is relevant to the study being a cross sectional study, so it did not include follow up or response to management, and we hope that future prospective studies will address these aspects. On the other hand, the strength of our study is that it showed that PH as defined by a TRV cut off point, and TRV as a continuous variable were both correlated to markers of hemolysis.

## Conclusions

In this study we found that the PH is not uncommon, and is often missed, among Iraqi patients with TM and its pathogenesis is more closely linked to chronic hemolysis than to iron overload. Early screening by echocardiography starting at the age of 10 years, followed by confirmation by RHC in selected cases should be underscored. The importance of timely transfusion and adequate chelation is of paramount importance. Moreover, it may be worthwhile to consider initiating a multicenter study to assess the value of various proposed therapeutic options in this part of the world.

## Supporting information

**S1 Table. The main clinical, laboratory and echocardiographic findings in each of 100 thalassemia major patients enrolled in the current study.**
(XLSX)

## Acknowledgments

We acknowledge the cooperation of Dr. Saad Younis, the epidemiologist in the University of Duhok, the staff of echocardiography laboratory with staff of thalassemia center and the enrolled patients and their families.

## Author Contributions

**Conceptualization:** Ameen M. Mohammad, Muna A. Kashmoola, Nasir Al-Allawi.

**Data curation:** Ameen M. Mohammad, Mohammed M. Dawad.

**Formal analysis:** Ameen M. Mohammad, Mohammed M. Dawad, Muna A. Kashmoola, Nasir Al-Allawi.

**Investigation:** Ameen M. Mohammad, Mohammed M. Dawad.

**Methodology:** Ameen M. Mohammad, Muna A. Kashmoola, Nasir Al-Allawi.

**Project administration:** Muna A. Kashmoola.

**Resources:** Mohammed M. Dawad.

**Supervision:** Ameen M. Mohammad, Muna A. Kashmoola, Nasir Al-Allawi.

**Validation:** Ameen M. Mohammad, Mohammed M. Dawad, Muna A. Kashmoola, Nasir Al-Allawi.

**Visualization:** Ameen M. Mohammad, Mohammed M. Dawad, Muna A. Kashmoola, Nasir Al-Allawi.

**Writing – original draft:** Ameen M. Mohammad, Mohammed M. Dawad, Muna A. Kashmoola, Nasir Al-Allawi.

**Writing – review & editing:** Ameen M. Mohammad, Muna A. Kashmoola, Nasir Al-Allawi.

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
