## [Decision Letter · Decision Letter 0]

8 Sep 2020

PONE-D-20-17752

Doppler-Defined Pulmonary Hypertension in β-Thalassemia Major in Kurdistan, Iraq

PLOS ONE

Dear Dr. Mohammad,

Thank you for submitting your manuscript to PLOS ONE. After careful consideration, we feel that it has merit but does not fully meet PLOS ONE’s publication criteria as it currently stands. Therefore, we invite you to submit a revised version of the manuscript that addresses the points raised during the review process.

This is an interesting study investigating cardiopulmonary complications of thalassemia. Authors were especially interested in PH, finding that PH is not uncommon in Iraqi patients with thalassemia major.

Although the manuscript is of the interested of cardiology community, reviews raised several points requiring careful revision.

We look forward to receiving your revised manuscript.

Kind regards,

Otavio Rizzi Coelho-Filho, M.D., Ph.D., M.P.H.

Academic Editor

PLOS ONE

Additional Editor Comments:

This is an interesting study investigating cardiopulmonary complications of thalassemia. Authors were especially interested in PH, finding that PH is not uncommon in Iraqi patients with thalassemia major.

Although the manuscript is of the interested of cardiology community, reviews raised several points requiring careful revision.

Journal Requirements:

Reviewers' comments:

Reviewer's Responses to Questions

**Comments to the Author**

1. Is the manuscript technically sound, and do the data support the conclusions?

Reviewer #1: Yes

Reviewer #2: Yes

2. Has the statistical analysis been performed appropriately and rigorously? 

Reviewer #1: Yes

Reviewer #2: Yes

3. Have the authors made all data underlying the findings in their manuscript fully available?

Reviewer #1: Yes

Reviewer #2: Yes

4. Is the manuscript presented in an intelligible fashion and written in standard English?

Reviewer #1: Yes

Reviewer #2: Yes

5. Review Comments to the Author

Reviewer #1: The trial is important to better understand thalassemia and why is very important to make the correct treatment.

Please, inform in the introduction that PH is mainly in NTDT

At the end of introduction, be more objective when describing the trial objective

PLOS uses the reference style outlined by the International Committee of Medical Journal Editors (ICMJE), also referred to as the “Vancouver” style. Please, correct the references.

Please see some suggestions in the attached PDF.

English could be improved.

Reviewer #2: Despite the limitations of echocardiography to evaluate RV function, it is possible by the FAC (fractional area change) and TAPSE ( tricuspid annular plane systolic excursion). So if you have these data, I think it is important to add it to the manuscript, at least for the patients with pulmonary hypertension. It is not essential because it is not one of the objectives but it is important.

6. PLOS authors have the option to publish the peer review history of their article (what does this mean?). If published, this will include your full peer review and any attached files.

Reviewer #1: No

Reviewer #2: No

---

## [Author Response · Author response to Decision Letter 0]

1 Nov 2020

RESPONSE TO EDITORS & REVIEWERS

Journal Requirements:

Style requirements for title page, abstract, references were followed and appropriate corrections were made where needed.

Added at the appropriate site

Reviewers' comments:

Reviewer's Responses to Questions

Comments to the Author

1. Is the manuscript technically sound, and do the data support the conclusions?

Reviewer #1: Yes

Reviewer #2: Yes

2. Has the statistical analysis been performed appropriately and rigorously?

Reviewer #1: Yes

Reviewer #2: Yes

3. Have the authors made all data underlying the findings in their manuscript fully available?

Reviewer #1: Yes

Reviewer #2: Yes

4. Is the manuscript presented in an intelligible fashion and written in standard English?

Reviewer #1: Yes

Reviewer #2: Yes

5. Review Comments to the Author

Reviewer #1: The trial is important to better understand thalassemia and why is very important to make the correct treatment.

Please, inform in the introduction that PH is mainly in NTDT

A statement to this effect was introduced: lines 55-59 page 3 First paragraph introduction.

At the end of introduction, be more objective when describing the trial objective

The statement at the end of introduction was modified as requested: Lines 69-70 page 3 introduction.

PLOS uses the reference style outlined by the International Committee of Medical Journal Editors (ICMJE), also referred to as the “Vancouver” style. Please, correct the references.

The reference style used by the Journal was employed as requested pages 13-16

Please see some suggestions in the attached PDF.

Almost all suggested modifications were performed as requested by reviewer, and we are very thankful for the reviewer's constructive and in depth review.

The modifications include changing two references, and changing some statements as requested.

English could be improved.

Multiple changes were introduced as appropriate.

Reviewer #2: Despite the limitations of echocardiography to evaluate RV function, it is possible by the FAC (fractional area change) and TAPSE ( tricuspid annular plane systolic excursion). So if you have these data, I think it is important to add it to the manuscript, at least for the patients with pulmonary hypertension. It is not essential because it is not one of the objectives but it is important.

We agree with the reviewer of the additional informative value of FAC and TAPSE, unfortunately this information is not available to us to add to the results.

---

## [Decision Letter · Decision Letter 1]

25 Nov 2020

Doppler-Defined Pulmonary Hypertension in β-Thalassemia Major in Kurdistan, Iraq

PONE-D-20-17752R1

Dear Dr. Mohammad,

We’re pleased to inform you that your manuscript has been judged scientifically suitable for publication and will be formally accepted for publication once it meets all outstanding technical requirements.

Kind regards,

Otavio Rizzi Coelho-Filho, M.D., Ph.D., M.P.H.

Academic Editor

PLOS ONE

Additional Editor Comments (optional):

Authors have satisfactory responded all raised questions.

Reviewers' comments:

Reviewer's Responses to Questions

**Comments to the Author**

1. If the authors have adequately addressed your comments raised in a previous round of review and you feel that this manuscript is now acceptable for publication, you may indicate that here to bypass the “Comments to the Author” section, enter your conflict of interest statement in the “Confidential to Editor” section, and submit your "Accept" recommendation.

Reviewer #1: All comments have been addressed

2. Is the manuscript technically sound, and do the data support the conclusions?

Reviewer #1: Yes

3. Has the statistical analysis been performed appropriately and rigorously? 

Reviewer #1: Yes

4. Have the authors made all data underlying the findings in their manuscript fully available?

Reviewer #1: Yes

5. Is the manuscript presented in an intelligible fashion and written in standard English?

Reviewer #1: Yes

6. Review Comments to the Author

Reviewer #1: Study accepted for publication. I hope it helps in the care of the patient with thalassemia! Congratulations!

7. PLOS authors have the option to publish the peer review history of their article (what does this mean?). If published, this will include your full peer review and any attached files.

Reviewer #1: No

---

## [Editor Report · Acceptance letter]

1 Dec 2020

PONE-D-20-17752R1 

Doppler-Defined Pulmonary Hypertension in β-Thalassemia Major in Kurdistan, Iraq 

Dear Dr. Mohammad:

I'm pleased to inform you that your manuscript has been deemed suitable for publication in PLOS ONE. Congratulations! Your manuscript is now with our production department. 

Kind regards, 

on behalf of

Dr. Otavio Rizzi Coelho-Filho 

Academic Editor

PLOS ONE